# Magnetic Resonance Volumetric Quantification of Vestibular Endolymphatic Hydrops in Patients with Unilateral Definite Meniere’s Disease Using 3D Inversion Recovery with Real Reconstruction (3D-REAL-IR) Sequence

**DOI:** 10.3390/jcm12185965

**Published:** 2023-09-14

**Authors:** Víctor Suárez-Vega, Raquel Manrique-Huarte, Pablo Dominguez, Melissa Blanco, Alberto Alonso-Burgos, Nicolás Pérez-Fernández

**Affiliations:** 1Department of Radiology, Clinica Universidad de Navarra, 28027 Madrid, Spain; vvega@unav.es; 2Department of Otorhinolaryngology, Clinica Universidad de Navarra, 31008 Pamplona, Spain; rmanrique@unav.es; 3Department of Radiology, Clinica Universidad de Navarra, 31008 Pamplona, Spain; pdaniel@unav.es; 4Department of Otorhinolaryngology, Clinica Universidad de Navarra, 28027 Madrid, Spain; melimelo15@gmail.com (M.B.); nperezfer@unav.es (N.P.-F.)

**Keywords:** Ménière’s disease, endolymphatic hydrops, 3D-REAL-IR, vestibular endolymphatic ratio

## Abstract

Background: The 3D-REAL-IR MRI sequence allows for an in vivo visualization of endolymphatic hydrops. Qualitative assessment methods of the severity of vestibular and cochlear hydrops are the most commonly used. Methods: A quantitative volumetric measurement of vestibular EH in patients with definite unilateral Ménière’s disease using the 3D-REAL-IR sequence and the calculation of the endolymphatic ratio (ELR) was intended. Results: Volumetric calculations of the vestibules, vestibular endolymph and vestibular ELR are performed in 96 patients with unilateral Ménière’s disease and correlated with classic qualitative grading scales. Conclusions: Quantitative volumetric measurement of vestibular hydrops using the 3D-REAL-IR sequence is feasible and reproducible in daily clinical practice. Vestibular ELR values exceeding 60% defined radiologically significant vestibular hydrops, while values below 30% defined radiologically non-significant vestibular hydrops.

## 1. Introduction

Ménière’s disease (MD) is a chronic condition of the inner ear of unknown etiology, likely multifactorial, with a prevalence ranging from 3.5 to 500 cases per 100,000 inhabitants [1].

It affects both the organs of hearing and balance, and is characterized by a triad of episodes of vertigo attacks, fluctuating hearing loss, and tinnitus. Patients also often report a sensation of ear fullness, gait disturbances, postural instability, nausea, and “drop attacks.” The combined effect of genetic and environmental factors likely determines the onset of the disease [2]. Therefore, MD is one of the fundamental differential diagnoses to consider in any patient who presents with repeated vertigo attacks without a known trigger. Although initially the disease shows a pattern of relapses with subsequent complete recovery, over time, deficits in both hearing and balance become permanent [3].

First thought to represent a brain disorder, in 1861 Prosper Ménière claimed the inner ear as the anatomic location for this syndrome [4]. The endolymphatic hydrops (EH) is a condition characterized by a distention of the structures of the inner ear containing endolymph and it is generally accepted as the pathologic counterpart. An increase in volume within these structures is tightly correlated with the symptoms of MD, as observed in temporal bone analysis [5].

So far, the role of Magnetic Resonance Imaging (MRI) was limited to rule out secondary causes mimicking MD.

The imaging basis of EH relies on the fact that gadolinium-based MR contrast diffuses to the perilymph—but not to the endolymph—altering the perilymph signal and allowing later discrimination between both components [6].

The work of Naganawa et al. [7], lays the groundwork for the study of EH through a 3 Tesla MRI using a specifically tailored MRI sequence. That study had three main objectives: (1) to attempt to determine the optimal time for image acquisition using a 3D fluid-attenuated inversion recovery (3D-FLAIR) sequence after a single intravenous injection of Gd at a dose of 0.1 mmol/kg; (2) to assess the quality of enhancement in healthy volunteers; and (3) to confirm whether this sequence could separately distinguish perilymph and endolymph. The interesting contribution of this study was the use of FLAIR sequences instead of T1-weighted sequences for enhancement evaluation. In fact, they made a comparison between the T1 and 3D-FLAIR sequences using the same Gd dose. While the T1 sequences were unable to detect cochlear enhancement, the FLAIR sequences were much more sensitive in detecting this perilymphatic enhancement.

Later on, they described another sequence named 3D inversion-recovery (3D-IR) turbo spin echo with real reconstruction, showing higher contrast between the non-enhanced endolymph and the surrounding bone, thanks to a shorter inversion time than the FLAIR sequence [8].

Nowadays, two main sequences (3D-FLAIR and “Inversion Recovery with REAL reconstruction” (3D-REAL-IR)), and two methods of contrast administration (intravenous or intratympanic) have been consolidated [9].

In a recent work from our group comparing both sequences [10], we found that the 3D-REAL-IR sequence exhibits a higher signal-to-noise ratio than the 3D-FLAIR.

In addition, a significantly higher rate of cochlear EH detection is observed in the 3D-REAL-IR sequence compared to 3D-FLAIR, especially in cases of mild cochlear EH that are classified as normal in 3D-FLAIR. The rates of vestibular EH detection do not differ as much when comparing both sequences, although there does appear to be a certain tendency towards the superiority of the 3D-REAL-IR sequence.

Regarding the severity grading of EH, visual, qualitative, or semi-quantitative scales have been the most widely used from the beginning (largely due to their greater ease, reproducibility, and applicability in daily clinical practice). One of its main limitations is that it actually assesses the degree of distention of the endolymphatic space in a single cross-section, often the plane that passes through the horizontal semicircular canal (HSC), referred to as “midmodiolar” level.

Quantitative volumetric EH assessment, which might be assumed conceptually as more accurate than qualitative assessment, is much less widespread due to different reasons. First, reliable volumetric measurement of structures as small as the vestibule or cochlear canal is difficult to perform, and there is no widely accepted and standardized method, as it heavily relies on the software used. Second, semi-quantitative or automated volumetric methods exist in other areas of radiology but their application in such small anatomical structures often results in measurement errors. Third, the manual delineation of the boundaries of the endolymph in each of the slices of the volumetric sequence, that is the most commonly used technique, requires greater effort and is a time-consuming task.

The main purposes of this study were:

To perform a quantitative volumetric measurement of vestibular EH in patients with definite unilateral Ménière’s disease using a specific optimized sequence (3D-REAL-IR) in a semi-automated manner, conducted retrospectively and cross-sectionally, aiming for feasibility and reproducibility in daily clinical practice.

To calculate the percentages of vestibular EH involvement (endolymphatic ratio or ELR) and compare these percentages with visual qualitative/semi-quantitative scales by using absolute volumetric measurements of vestibular volume and endolymph volume.

To attempt to define threshold values of radiologically significant EH in correlating ELRs with grades from semi-quantitative scales.

## 2. Materials and Methods

### 2.1. Patients

The Research Ethics Committee of the University of Navarra (project number 2021.199) approved this study. All patients included in this study provided explicit consent for the use of their data for research purposes and written informed consent was obtained from all subjects.

From September 2016 until June 2022, 96 patients diagnosed with MD according to the Bárány criteria [11] were referred to our clinic for MR hydrops imaging. All the patients were evaluated as to fulfill the criteria for diagnosis of definite unilateral disease. Demographic data included the disease duration, number of vertigo attacks during the 6-months period before being seen and days since the last vertigo spell. A complete otological and otoneurological assessment was performed at bedside. Audiometry (pure tone) and vestibular testing (video-head impulse test and vestibular evoked myogenic potentials) were also performed.

All MR studies were performed using 3 Tesla magnets, either a Magnetom Vida or a Magnetom Skyra (Siemens Healthineers, Erlangen, Germany) with 20-channel and 32-channel phased-array receiver coils, respectively, and with patients lying in a supine position. A single dose of intravenous paramagnetic contrast agent gadobutrol 0.1 mmol/mL (Gadovist, Bayer AG, Zurich, Switzerland) was administered at the manufacturer’s recommended dose of 0.1 mL per kg of patient’s body weight. Images were acquired after a 4 h interval following the contrast administration.

### 2.2. MRI Hydrops Sequences

The imaging protocol consisted of:–A heavily T2 weighted sequence also referred to as “cysternography sequence” (T2 3D Sampling Perfection with Application optimized Contrasts using different flip angle Evolution (SPACE)) with the following parameters: section thickness, 0.5 mm; TR, 1400 ms; TE, 152 ms; flip angle, 120°; bandwidth, 289 Hz/pixel; voxel size, 0.5 × 0.5 × 0.5; and scan time, 5 min.–3D-REAL-IR: section thickness, 0.8 mm; TR, 16,000 ms; TE, 551 ms; TI: 2700 ms; flip angle, 140°; bandwidth, 434 Hz/ pixel; voxel size, 0.5 × 0.5 × 0.8; and scan time, 11 min.

A detailed description of the sequence parameters can be found in Table 1.

### 2.3. Qualitative Hydrops Imaging Assessment

In the qualitative/semi-quantitative assessment of EH degree, both ears were evaluated, and the degree of vestibular and cochlear EH was separately recorded for each of the inner ears.

The degree of cochlear hydrops was determined using a three-level scale ranging from 0 to 2 (normal, mild, and severe), with the anatomical reference plane being the axial section passing through the modiolus and encompassing the greatest portion of the cochlea in that particular section as previously published [12,13].

For the assessment of vestibular hydrops, a four-level scale was used [14,15]. The optimal visualization plane was the one displaying the greatest anatomical extent of the vestibule, often including the plane of the horizontal semicircular canal (HSC).

### 2.4. Post-Processing of Images for Volumetric Calculations

The volumetric measurement of the entire vestibule (VV) was performed in a semi-automated manner using the cisternography sequence (T2 SPACE). Subsequently, the volume of vestibular endolymph (EndV) was measured using the 3D REAL-IR sequence.

In each patient, four different volumes were calculated, one for each of the following elements:–Volume of the right vestibule using the cisternography sequence.–Volume of the left vestibule using the cisternography sequence.–Volume of vestibular endolymph in the right ear using the 3D-REAL-IR sequence.–Volume of vestibular endolymph in the left ear using the 3D-REAL-IR sequence.

In total, four volumetric measurements per patient across 96 patients amount to 384 volumetric assessments.

All volumetric measurements were conducted using the advanced visualization software Siemens Syngo.via version VB50B (Siemens Healthineers). The same neuroradiologist (V. S.), with more than 10 years of experience, performed all measurements. The workflow consisted of:–Launching the MM Reading (MultiModality Reading) visualization workflow.–Selecting sequences (T2 SPACE cisternography and 3D REAL-IR).–Radiological window parameters for the 3D-REAL-IR sequence of C 38 and W 177 pixel intensity.–In each sequence, select the Multiplanar Reconstruction (MPR) mode as they open in 2D mode by default.–Select the advanced “Freehand VOI” tool.–Each of the inner ears were scanned with approximately 8 slices per sequence. In each of these images corresponding to a slice, using the computer mouse and the pencil tool, the neuroradiologist manually delineated the anatomical boundaries of the vestibule in the T2 SPACE sequence or the boundaries of the vestibular endolymph in the 3D REAL-IR sequence.–Once all slices were delineated, click on the total VOI button, and the program automatically generated a graph indicating the total volume of the sum of these delineations. The volume is measured in cm^3^.

In the anatomical delineation of the vestibular boundary volume, the ampullary dilatations of the semicircular canals (SCCs) were not included in the measurements until they were anatomically integrated into the boundaries of the vestibule.

Once the volumes of the vestibules and vestibular endolymph were obtained, the vestibular endolymphatic ratio (vestibular ELR) was calculated as a percentage using the following formula:EndVVV×100

Vestibular ELR was calculated for both the affected ear and the “healthy” or asymptomatic contralateral ear. Figure 1 depicts how calculations were made.

### 2.5. Statistical Analysis

The statistical analysis of the data was performed using SAS^©^ software v9.4, SAS Institute, Cary, NC, USA.

Description of quantitative values was performed using descriptive statistics including mean, standard deviation, and confidence interval for the mean. Since it is possible that data distributions do not follow a Gaussian distribution, robust statistics such as median, interquartile range, as well as minimum and maximum values are also provided. Distributions of categorical variables were described using absolute frequencies and percentage distributions. The graphical description of quantitative variables was performed using box and whisker plots.

To detect statistically significant relationships between categorical variables, the Chi-square test was used, or the Fisher’s exact test was applied in case none of the variables fulfilled the assumption for the Chi-square test (80% of the cells should have an expected frequency > 5).

To test for statistically significant differences in continuous variables of ratio or interval scale between groups, the Student’s *t*-test, or its extension, the Snedecor’s F test, were utilized. In case assumptions were not met, the Mann-Whitney U test or the Kruskal-Wallis test were applied.

Partitioning models and search for cut-off points were conducted using segmentation tree models. The goodness-of-fit measure for the model was the area under the ROC curve.

All tests were conducted two-tailed, and a significance level of 5% (*p* < 0.05) was used for hypothesis rejection.

## 3. Results

In this study, 96 patients with a confirmed diagnosis of unilateral definite Ménière’s disease were enrolled. Among them, 47 were male (49%) and 49 were female (51%). The mean age was 54.79 years (±11.54). The overall data of the patient cohort are summarized in Table 2.

In Table 3 we present the distribution of MRI results. Significant differences were observed between the “absent” and “severe” degrees in both affected and not affected ears, both at the vestibular and cochlear levels. When applying the Bowker symmetry test between cochleae and vestibules, a statistically significant difference is shown between the affected and not affected sides in both cases (*Χ^2^* 72.8095, *p* < 0.0001).

Regarding a purely morphological or anatomical assessment of the total vestibular volume, measurements were taken in the T2 cisternography sequence. When comparing absolute vestibular volumes between the left and right sides, or between the affected and unaffected sides, no significant differences were found. The mean was 0.08 cm^3^ ± 0.01, sign rank *p* = 0.7454.

When considering the vestibular volume of only the affected ears and attempting to find differences based on gender, no significant differences in the vestibular volume of the affected ears between males and females were observed. The unpaired Student’s *t*-test yielded *p* = 0.2476.

### 3.1. Comparative Analysis of the ELR

When comparing the EndV, statistically significant differences were found between the affected and unaffected ears. In the unaffected ears, there was a percentage reduction in volume of 35 ± 46.2% (mean ± SD). The means were 0.06 cm^3^ for the affected ears compared to 0.03 cm^3^ for the unaffected ears, nearly doubling the value, *p* < 0.0001.

Because of that, when calculating the ELRs and comparing between the symptomatic and asymptomatic sides, once again, we observed distinct differences, with significantly increased vestibular ELRs in the symptomatic ears (73.15% vs. 37.22%, sign rank test *p* < 0.0001).

When grading vestibular EH in symptomatic ears according to the qualitative classification of four levels (absent, mild, moderate, severe), and calculating the mean volumetric vestibular ELR in each of these four groups, statistically significant differences were obtained between all groups with *p* < 0.0001 according to the ANOVA test, except for the comparison between the basent-mild groups (*p* = 0.2814). These results are shown in Figure 2.

### 3.2. ELRs Threshold Values

In the search for cut-off points or threshold values, we defined the vestibular ELR percentage that optimally separated the patients who would have a radiologically significant degree of EH (moderate-severe grades) from those with mild or absent EH. Similarly, we defined a cut-off point at the lower end to locate the optimal value of non-radiologically significant EH. For these tasks, we employed the statistical technique of segmentation trees. Considering a vestibular ELR of 60% as the cut-off point for radiologically significant EH, the sensitivity was 88.7%, while its specificity was 82.1%. When plotting the results on an ROC curve, we obtained an area under the curve (AUC) of 0.9037, corresponding to a “very good” test for discriminating between significant and non-significant EH when using a 60% ELR cut-off point.

All these values are depicted in the chart in Figure 3.

Conversely, in the search for an optimal lower cut-off point that separated patients with “none-mild” scores from those with “moderate-severe” scores, focusing on radiologically “non-significant” or “less significant” vestibular EH, all the ELRs of unaffected vestibules were considered. Two cut-off points were generated: 30% and 55%. In other words, groups of patients with a vestibular ELR above 55%, between 30% and 55%, and below 30%. In this scenario, the value of AUC was 75.4%, as shown in the ROC curve. The probability of having ELR below 30% in the unaffected ear and a qualitative vestibular EH value of “absent” was 95.5%.

For results, see the chart in Figure 4.

## 4. Discussion

In 2015, Gürkov and colleagues [16] published the first article that demonstrated in vivo volumetric quantification of HE. It is one of the most important publications due to its relevance and methodology. The study included 16 patients with unilateral definite Ménière’s disease. Despite the small sample size, the results were robust and remain highly relevant today. The contrast administration route used was intratympanic, and the selected sequences were T2 cisternography for calculating the total VV and 3D-REAL-IR for EndV calculation. Absolute volumetric measurements in mm^3^ were obtained for both the cochlea and vestibule, as well as for the semicircular canals, along with three-dimensional representations. The authors emphasized the challenges in visualizing endolymph in the semicircular canals, and thus they did not perform endolymphatic volumetric measurements for them. They introduced the concept of ELR as an alternative approach.

The volumetric values from that article can serve as a guide for future studies. Of particular interest to us as a reference for comparison were the vestibular ELR, which in Gürkov’s work showed an average value of 28%, with a maximum of 40% and a minimum of 12%. These values were surprisingly low, considering that the population consisted of patients with unilateral definite Ménière’s disease. The authors themselves acknowledged this as one of the limitations of the study. They also found a significant correlation between the severity of cochlear endolymphatic hydrops and hearing loss.

Another significant article in the field of volumetric calculations comes from the Japanese group led by Hiroshi Inui [17]. This initial study encompassed two key aspects: (1) detailed description of the segmentation volumetric method that would be used in subsequent works by the group (KIIS method) and (2) as long as subjects were “healthy volunteers” without symptoms of Ménière’s disease, this study served as a reference guide for “normal values” concerning endolymphatic volumes.

That work not only contributed to the understanding of segmentation techniques but also offered a valuable baseline for normal EndV values in individuals without symptoms of Ménière’s disease.

That same year, Inui and colleagues [18] conducted an initial volumetric assessment using the T2 cisternography sequence, but without studying the endolymphatic component. Based on the hybrid images generated using the HYDROPS method, which are very similar to the 3D-REAL-IR sequence previously explained, they first manually delineated the endolymphatic and anatomical boundaries of the inner ear on the workstation console. Subsequently, they merge both volumes and calculate the ELR.

This methodology allowed them to explore the volumetric aspects of the inner ear, specifically the endolymphatic space, using a sequence that is akin to the 3D-REAL-IR sequence. This approach enables a comprehensive analysis of the endolymphatic volume and its relationship to anatomical structures.

Although the average vestibular ELR was 16.2%, they obtained a range of values spanning from 8% to 24%. This variability, which is also observed in our cohort, might stem from the different distribution patterns of endolymph in the vestibule, even in the absence of significant vestibular hydrops. The outcomes of Inui’s group’s study, similarly to the findings reported in our work, where the vestibular content is considered as a whole, demonstrated such variability in vestibular volumes. Studies that calculate vestibular ELR based on divisions of the areas of only one or two cuts tend to exhibit less variability in results.

An example of this method for calculating vestibular ELR by dividing restricted areas at a level of the vestibule can be found in the study by Liu and colleagues [19], conducted on both healthy volunteers and patients with Ménière’s disease. The mean vestibular ELR in patients with Ménière’s disease was 44%, whereas the mean vestibular ELR in healthy volunteers was 30%. Despite the volunteers being healthy, the variability in the amount of endolymph within these vestibules remains a consistent observation in the studies mentioned.

Subsequently, the group led by Inui and colleagues utilized the volumetric segmentation technique and the values from healthy volunteers they had previously reported to publish a study [20] in which they calculated volumetric values for different patient cohorts as well as healthy volunteers. The cohorts included patients with sudden deafness, patients with sensorineural hearing loss at low frequencies, and those with unilateral Ménière’s disease. Vestibular ELR were significantly higher in the Ménière’s disease cohort compared to the other groups.

Using the same volumetric quantification method, Ito and colleagues [21] replicated the results once again by comparing patients with unilateral Ménière’s disease (82 patients), bilateral Ménière’s disease (16 patients), and healthy volunteers (47 healthy volunteers). They found that indeed, the vestibular ELR in the affected ears of unilateral Ménière’s disease and in both ears of bilateral Ménière’s disease were significantly higher than in healthy volunteers.

Very recently, the same group of Inui and colleagues [22] validated their volumetric quantification method in terms of diagnostic precision for Ménière’s disease, emphasizing the added value of the presence of hydrops in the SCCs.

The publication by Homann and colleagues [23] once again emphasizes the active role of the specialist in actively delineating anatomical boundaries and the endolymphatic space.

We will highlight a less widespread volumetric method, previously mentioned and used by Peng and colleagues [24] for the volumetric calculation of vestibular hydrops. In this method, they measured the endolymphatic and perilymphatic area in various slices and multiplied them by the slice thickness of the sequence.

Furthermore, the role of Artificial Intelligence (AI) in volumetric quantification has been explored, with tools developed for automated segmentation of the endolymphatic space in the form of *pipelines*. Some of these methods are available in an open-source format, subject to consultation with the authors, in the form of a plug-in [25].

### 4.1. Volumetric Measurements of Our Patient Cohort

When measuring the total VV using the T2 cisternography sequence (T2 SPACE), we obtained an average value of 0.08 cm^3^. This mean remained consistent both when comparing sides (right versus left) and when comparing symptomatic ears with asymptomatic ears. This volumetric value aligns precisely with the vestibular volumetric value reported by Gürkov, which had an average of 0.08 cm^3^ [16].

In subsequent articles that calculated vestibular volumetric values, similar results to ours were also obtained. In the work by Inui and colleagues [17], which included patients with Ménière’s disease and chronic rhinosinusitis (considered a healthy control group), VV values of 0.07 cm^3^ were obtained in patients without Ménière’s disease, and a volume very close to 0.08 cm^3^ (0.0767 cm^3^) was observed in ears of patients with unilateral Ménière’s disease. The same group, in another article [20] that included various patient groups with diseases related to endolymphatic hydrops (unilateral Ménière’s disease, “cochlear” Ménière’s disease, sudden low-tone sensorineural hearing loss), obtained very similar vestibular volume values. In another publication involving 100 healthy volunteers [26] from the same group of authors, the VV again hovered around 0.07 cm^3^.

In our cohort of patients, when comparing the absolute EndV obtained using the REAL-IR sequence, we found statistically significant differences between the affected and unaffected ears (symptomatic vs. asymptomatic). The mean EndV values in affected ears were three times higher than those in asymptomatic ears (means of 0.03 cm^3^ in affected ears and 0.01 cm^3^ in unaffected ears). If we examine the maximum and minimum values, in affected ears, we observed maximum values of 0.14 cm^3^ and minimum values of 0.01 cm^3^, while in unaffected ears, the maximum values reached 0.07 cm^3^ and the minimum values were 0.01 cm^3^. Notably, there was significant overlap in the minimum values of both groups.

Comparing these values with those reported in the literature, articles measuring EndV in patients without symptoms of Ménière’s disease obtained values of 11.5 ± 7.1 mm^3^ [17], falling within the lower range of volumes for asymptomatic ears in our cohort. Similarly, in the various patient groups included in another article [20], the mean EndV values in patients with unilateral Ménière’s disease were 0.024 cm^3^, very close to our mean value of 0.03 cm^3^. The results from Gürkov’s group [16] also aligned, with a mean EndV value of 0.022 cm^3^, slightly lower. Therefore, despite different methodologies, the reviewed articles yielded very similar volumetric results, considering the small volumetric magnitudes involved, where slight differences were magnified.

In Table 4, a summary is provided that compiles various published values from the literature for total VV, EndV, and vestibular ELR. A comparison is also made with the average values in our cohort.

### 4.2. Volumetric Measurements (ELR) of Our Patient Cohort

Based on the data presented so far, isolated absolute volume values, whether of perilymph or endolymph, do not provide us with an accurate understanding of the degree of involvement. This is the reason for using values such as the vestibular ELR.

In the affected ears, the vestibular ELR values had a mean of 73.15%, with a median of 66%. The maximum value obtained of 155% indicated that the endolymphatic volume greatly exceeds the limits of the vestibule, possibly herniating into the SCCs. However, in the group of affected ears, there were also minimum values as low as 11%. This means that even in patients with unilateral definite MD, there were cases where the symptomatic ear did not exhibit vestibular hydrops at the time of the test. Conversely, in the asymptomatic unaffected ears, the mean vestibular ELR was 37%, with a median of 37% (indicating more homogeneous values in this group). Nevertheless, there were extreme maximum values as high as 88% (from an individual patient) and minimum values as low as 9%. In other words, even in patients with unilateral MD, the unaffected ear could show vestibular ELR values within the range considered pathological. However, despite these extreme values (considered “outliers”) all differences between the two groups were statistically significant by a wide margin.

When evaluating the vestibular ELR in the affected and unaffected ears, classified according to the qualitative grading of vestibular EH with four degrees, we obtained mean vestibular ELR values for each group. Specifically, the mean vestibular ELR was 31.08% in the “no EH” group, 46.92% in the “mild” group, 67.32% in the “moderate” group, and 104.4% in the “severe” group. As the degree of vestibular hydrops increased, the mean vestibular ELR values also increased. Despite the overlapping values, particularly in the “no HE” and “mild” groups, all differences between the various groups were statistically significant, excluding the “no HE” to “mild” comparison.

When comparing our vestibular ELR values with those published by Ito and colleagues [21], we found the following considerations:

The values reported by Ito and colleagues for vestibular measurements in healthy control patients were 17.7 ± 10.2%. Our closest match to this would be the mean vestibular ELR of the “absent HE” group in unaffected ears, where we obtained values of 34.11 ± 10.2%. The vestibular ELR values for the affected ears in the “absent EH” group were 31.08 ± 13.73%. Both of these values were noticeably higher than those of the healthy controls.

The values reported for asymptomatic ears of patients with unilateral Ménière’s disease were 20.3 ± 14.6%, and for symptomatic ears, they were 37.5 ± 23.0%. In our cohort, the vestibular ELR values for asymptomatic ears were 37.22 ± 12.72%, and for symptomatic ears, they were 73.15 ± 31.78%.

In summary, our vestibular ELR generally showed higher values compared to the values reported by Ito and colleagues, both in healthy controls and in patients with Ménière’s disease, indicating a greater presence of endolymphatic hydrops in our cohort.

However, the results from the study by Gürkov and colleagues [16] were striking, as they included patients with definite unilateral Ménière’s disease, and the mean and maximum values of vestibular ELR were 28% and 40%, respectively. These values are much lower than expected when compared to our results. The difference in the number of patients in the samples (16 patients in Gürkov’s study, 96 in ours) could be one of the contributing factors.

### 4.3. Search for the Cutoff Point of Radiologically Significant Hydrops Using the Vestibular ELR Indicator

We aimed to identify a threshold value of vestibular ELR that indicated the boundary above which we can consider the vestibular hydrops radiologically significant. Conversely, we also intended to determine a threshold value below which we are likely dealing with a study showing vestibular hydrops that is radiologically non-significant.

In the visual assessment of vestibular hydrops, both the three-grade Baráth scale [13] and the modified four-grade Bernaerts scale [14] defined “moderate to severe” involvement when the amount of endolymph exceeded 50% of the vestibular surface in a single cross-section. Other authors have applied arbitrary divisions of the grades, as seen in the study by Homann [23], where the author assigned arbitrary cut-off points to correlate with volumetric values: a “no hydrops” grade below 33%, a “mild to moderate” grade between 33% and 50%, and a “severe” grade above 50%. However, the rationale for choosing these cut-off points was not provided.

Initially, we focused solely on the symptomatic or affected ears in order to create two dichotomous groups: patients classified with qualitative grades of “none-mild” versus patients with qualitative grades of “moderate-severe.” We employed a statistical tool called a segmentation tree. Upon completion, the analysis suggested that a vestibular ELR threshold of 60% was the optimal point to distinguish radiologically significant vestibular hydrops from radiologically non-significant cases. When conducting logistic regression to generate the ROC curve, the sensitivity at this threshold was 88.7%, and the specificity was 82.1%. The area under the ROC curve using this threshold would be 0.9037.

Similarly, we have sought to establish a threshold of vestibular ELR for radiologically non-significant hydrops. To achieve this, we selected the group of ears that were less likely to be affected by hydrops. Therefore, by focusing exclusively on the “non-affected” group of ears, we aimed to identify cutoff points that dichotomously differentiated: firstly, qualitative groups of “none-mild” versus “moderate-severe,” and secondly, we would repeat the calculations considering the segregation of “none” versus “mild” groups.

The result was that the cutoff point that exhibits an optimal discriminatory value for radiologically non-significant hydrops in our patient cohort was a vestibular ELR value of 30%.

However, there are some limitations in our study. Firstly, volumetric calculations have been limited to the vestibule. Although the same operator with extensive experience in inner ear anatomy performed all measurements, the influence of human factor in the intra-operator measurement error has not been addressed. Trying to avoid this situation, some studies involve multiple operators [22] and conduct measurements multiple times [21]. We also lack “normal” values from “healthy volunteers”, thus we need to refer to values published in the literature.

## 5. Conclusions

We can establish the following key findings from this study:

Quantitative volumetric measurement of vestibular hydrops using the 3D-REAL-IR sequence in a semi-automated manner through commercial software is feasible and reproducible in daily clinical practice.

Vestibular ELR is the quantitative parameter that best represents the percentage of vestibular hydrops involvement and is the most widely used and recognized in the literature.

Vestibular ELR values exceeding 60% define radiologically significant vestibular hydrops, while values below 30% define radiologically non-significant vestibular hydrops.

## Figures and Tables

**Figure 1 jcm-12-05965-f001:**
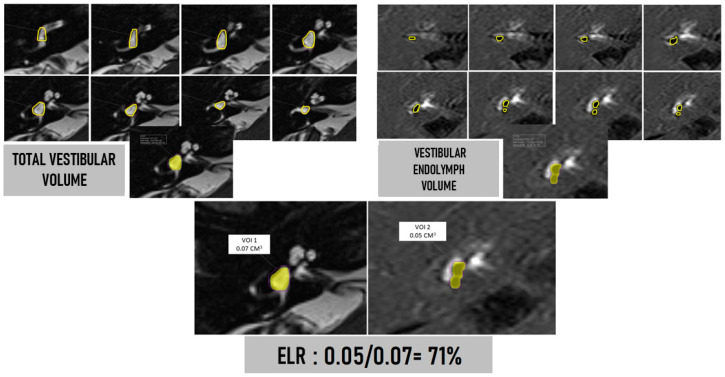
Upper left: T2 SPACE cisternography sequence. Manual delineation using the computer mouse of the anatomical boundaries of the vestibule in each of the slices containing it (yellow line). Once all contours were completed, the software itself calculated the total volume (last image below) and labeled it as VOI (volume of interest). The ampullary dilatations of the semicircular canals (SCCs) were not included in the measurements of total vestibular volume. Upper right: three-dimensional REAL-IR sequence. Manual delineation using the visualization console and computer mouse of the vestibular endolymph in each slice. Note that in this patient, the saccule has a larger volume than the utricle, but they do not merge (corresponding to a mild degree on the four-level vestibular hydrops scale). Volumes of the saccule and utricle were separately outlined, although the software displayed the total volume as a sum of both. Bottom: calculation of the endolymphatic ratio (ELR) corresponding to the percentage of the total vestibular volume occupied by endolymph. In this patient, the saccule and utricle were visualized independently, although the saccule has a larger volume than the utricle. This corresponded to a mild degree of vestibular hydrops on the qualitative/semi-quantitative visual scale of four grades. However, upon performing volumetric measurement and considering the entire volume of endolymph (not just in that particular slice), the ELR provides us with a value of 71%.

**Figure 2 jcm-12-05965-f002:**
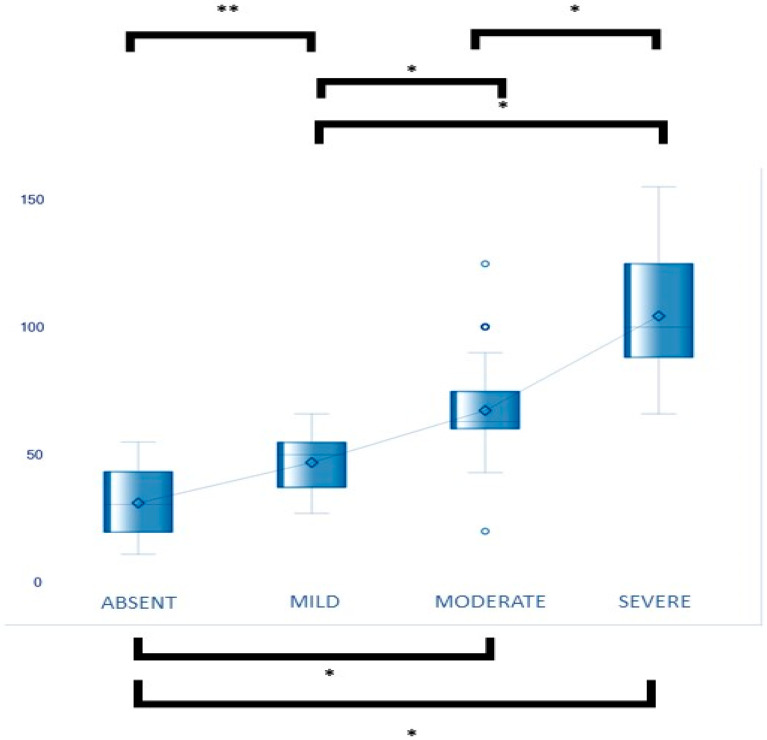
The mean ELRs are grouped according to the qualitative EH grades. The only comparison where there are no significant differences is between the absent-mild groups. In all other comparisons between groups, there are statistically significant differences. * ANOVA *p* < 0.001 ** ANOVA *p* = 0.2814 not significant.

**Figure 3 jcm-12-05965-f003:**
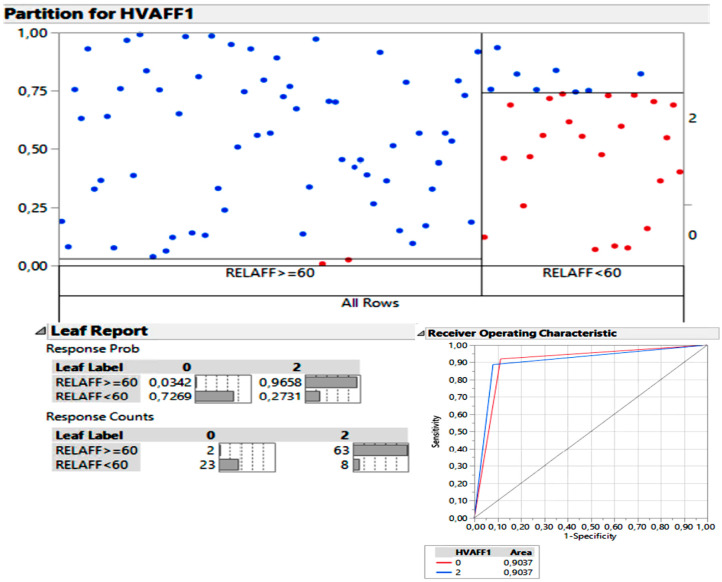
The segmentation tree suggested that ELR of 60% was the optimal threshold. When the score was higher than ELR of 60%, 96.5% of cases were “moderate-severe” (group 2 in the chart, Response Probability, corresponding to 63 blue dots in the chart, Response Counts). When the score was ELR less than 60%, the percentage of “absent-mild” patients was 72.6% (group 0 in the chart, Response Probability, 23 red dots in the chart, Response Counts). By using a ROC curve, we obtained the sensitivity and specificity for the 60% cut-off point, resulting in an AUC of 0.9037.

**Figure 4 jcm-12-05965-f004:**
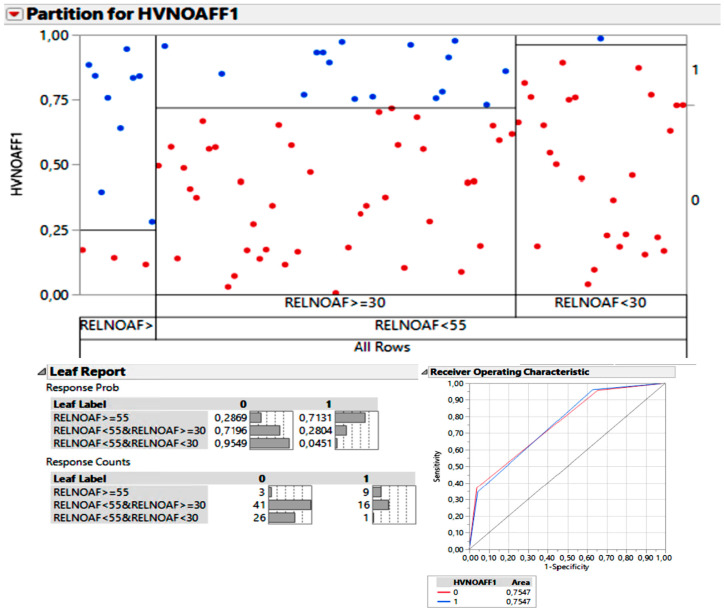
Two cut-off points were generated: 30% and 55%. The probability of having ELR below 30% in the unaffected ear and a qualitative vestibular EH value of “absent” was 95.5% (corresponding to the 26 red dots in the chart above the “RELNOAF < 30”, leaving only one blue dot classified as a qualitative vestibular EH of “moderate-severe”). In this scenario, the value of the AUC was 75.4%, as shown in the ROC curve.

**Table 1 jcm-12-05965-t001:** Parameters of the sequences.

Sequence	T2 SPACE	3D IR
Slice thickness (mm)	0.5	0.8
Slices	56	112
Field of view (mm)	160 × 160	134 × 200
Resolution (pixels)	320 × 320	259 × 384
Voxel size (mm)	0.5 × 0.5 × 0.5	0.5 × 0.5 × 0.8
TR (ms)	1400	16,000
TE (ms)	152	551
TI (ms)	N/A	2700
Flip angle	120	140
Bandwidth (Hz/Px)	289	434
Length (min:s)	4:44	10:56

TR: repetition time; TE: time to echo; TI: inversion time.

**Table 2 jcm-12-05965-t002:** Overall data of 96 patient’s cohort.

Variable	N	Mean (SD)	Min	Max	Median
Age	96 (100%)	54.79 (11.54)	28.00	75.00	55.50
Gender -Male -Female	47 (49%) 49 (51%)	N/A	N/A	N/A	N/A
Disease duration (years)	96 (100%)	5.81 (7.62)	0.00	40.00	3.00
First symptom -Synchronous -Hearing loss -Vertigo	35 (36%) 41 (43%) 20 (21%)	N/A	N/A	N/A	N/A
First consultation -First vertigo -Vertigo -Fluctuating HL -Instability -Progressive HL	4 (4%) 75 (78%) 6 (6%) 6 (6%) 5 (5%)	N/A	N/A	N/A	N/A
Days since last crisis	91 (95%)	44.73 (83.92)	1.00	365.0	15.00
Crisis last 6 months	92 (96%)	5.91 (5.30)	0.00	32	5
Migraine -No migraine -Migraine -Tension-type	80 (84%) 9 (9%) 6 (6%)	N/A	N/A	N/A	N/A

HL: hearing loss; N/A: not available.

**Table 3 jcm-12-05965-t003:** Depiction of frequency distributions of patients classified according to the qualitative scale of cochlear HE (three grades) and vestibular EH (four grades).

HE	N (%)	EH	N (%)
Cochlear affected -Absent -Moderate -Severe	18 (19%) 30 (31%) 48 (50%)	Cochlear not affected -Absent -Moderate -Severe	85 (89%) 10 (10%) 1 (1%)
Vestibular affected -Absent -Mild -Moderate -Severe	12 (12%) 13 (14%) 37 (39%) 34 (35%)	Vestibular not affected -Absent -Mild -Moderate -Severe	70 (73%) 18 (19%) 7 (7%) 1 (1%)

**Table 4 jcm-12-05965-t004:** Summary table of published volumetric values in the literature and comparison with our values.

	Gürkov et al. (2015) [15]	Homann et al. (2015) [23]	Inui et al. (2016) [16]	Inui et al. (2019) [19]	Inui et al. (2021) [21]	This Study Symptomatic	This Study Asymptomatic
VESTIBULAR VOLUME	0.08 cm^3^	1.62 cm^3^ (including SCCs)	0.075 cm^3^	0.07 cm^3^	0.07 cm^3^	0.08 cm^3^	0.08 cm^3^
VESTIBULAR ENDOL VOL	0.02 cm^3^	N/A	N/A	0.024 cm^3^	12.2 ± 7.5 mm^3^	0.06 cm^3^	0.03 cm^3^
VESTIBULAR ELR (mean, min and max)	28% 12–40%	N/A 12–80%	N/A	35.7%	20%	73% 11–155%	37% 9–88%
NUMBER PATIENTS MD/ HEALTHY	16/0	11/2	N/A	72/47	0/100	96	96

N/A: not available.

## Data Availability

Data is unavailable due to privacy or ethical restrictions.

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
