# Peer review of "Magnetic Resonance Volumetric Quantification of Vestibular Endolymphatic Hydrops in Patients with Unilateral Definite Meniere’s Disease Using 3D Inversion Recovery with Real Reconstruction (3D-REAL-IR) Sequence"

_jcm, 2023, doi:10.3390/jcm12185965_

Round 1

Reviewer 1 Report

The paper aims to measure vestibular endolymph hydrops in patients with definite unilateral Ménière's disease using 3D-REAL-IR sequence and calculating the endolymphatic ratio in 96 patients.

Areas of strength: qualified researchers and statistical analysis included in the paper.

Weaknesses: graphics (Fig. 3 and 4) and insufficient description of the measurement errors. The influence of the human factor in the measurements should be discussed.

Comments:

5- Recon-Struction  (Reconstruction)

161- Was this section thickness enough to achieve good accuracy?

175-177 – repeated 173-175

193- If the same experienced person made the measurements four times, what were the discrepancies for a single selected case?

222 – what does it mean ‘special care’? How exactly were the surfaces between dillatations and the vestibule determined?

309- Table 3 a typo : coclhear in two places

371 – vestibularELR (glued)

372, 373 – (s)ensitivity, (s)pecificity

404 – lower part of Fig 3 is cut and the description in the caption is unclear.

448 – lower part of Fig 4 is cut and the description in the caption is unclear.

468, 500, 507, …, and below  – vestibularELR (glued)

608- a typo ESTIBULAR  in Table 4 

608- use only one unit: cm3  or mm3.

623, 629 – (u)nilateral (d)efinite ?

658-  vestibular vestibularELR

some typos and missing spaces in the text.

Generally past tense should be used.

Reviewer 2 Report

Review Report

This paper presents a quantitative MRI method to measure endolymphatic hydrops in patients with Meniere's disease. 96 patients with unilateral definite Meniere's underwent MRI scans using specialized sequences. The vestibular endolymphatic space was manually segmented to calculate volumes.The endolymphatic ratio (ELR) was computed as the percentage of endolymph volume relative to total vestibular volume. 

 ELR was significantly higher in affected ears (mean 73%) compared to unaffected ears (mean 37%). The authors found that ELR correlated with subjective gradings of hydrops severity. An ELR threshold of 60% gave sensitivity of 89% and specificity of 82% for detecting significant hydrops. An ELR below 30% indicated radiologically non-significant hydrops.

Further, this method allows quantitative measurement of hydrops and could aid diagnosis. 

Demerits of this paper

 Manual segmentation is time-consuming but commercially available software enables feasibility.

 Study is limited by lack of healthy control data and single image analyst.

Demerits are the retrospective design and lack of healthy volunteer comparisons.

Merits of this paper

Merits include rigorous methodology, adequate sample size, and clinically relevant findings.  

Conclusion

 Overall this is a well-conducted study demonstrating a new quantitative imaging approach for hydrops.
